# On-Surface Synthesis of *sp*-Carbon Nanostructures

**DOI:** 10.3390/nano12010137

**Published:** 2021-12-31

**Authors:** Lina Shang, Faming Kang, Wenze Gao, Zheng Zhou, Wei Xu

**Affiliations:** Interdisciplinary Materials Research Center, College of Materials Science and Engineering, Tongji University, Shanghai 201804, China

**Keywords:** *sp*-carbon, on-surface synthesis, scanning tunneling microscope

## Abstract

The on-surface synthesis of carbon nanostructures has attracted tremendous attention owing to their unique properties and numerous applications in various fields. With the extensive development of scanning tunneling microscope (STM) and noncontact atomic force microscope (nc-AFM), the on-surface fabricated nanostructures so far can be characterized on atomic and even single-bond level. Therefore, various novel low-dimensional carbon nanostructures, challenging to traditional solution chemistry, have been widely studied on surfaces, such as polycyclic aromatic hydrocarbons, graphene nanoribbons, nanoporous graphene, and graphyne/graphdiyne-like nanostructures. In particular, nanostructures containing *sp*-hybridized carbons are of great advantage for their structural linearity and small steric demands as well as intriguing electronic and mechanical properties. Herein, the recent developments of low-dimensional *sp*-carbon nanostructures fabricated on surfaces will be summarized and discussed.

## 1. Introduction

Over the centuries, carbon allotropes have been playing a significant role in material science and engineering due to their outstanding electric, magnetic, optical, and mechanical properties. Compared to traditional carbon materials such as diamond and graphite, carbon nanomaterials consist of carbons with different hybridization types, namely linear (*sp*), planar (*sp*^2^), or tetrahedral (*sp*^3^) bond configurations, or even a combination of several types [1]. The investigation of novel carbon nanomaterials has attracted more attention since the first successful separation of monolayer graphene by Geim and Novoselov in 2004 [2]. In view of the remarkable semimetallic property, graphene can be modified to open a bandgap by heterogeneous element doping [3,4] and multifunctional nanoporous graphene [5,6]. Over the past decades, many breakthroughs for the fabrication of novel nanostructures have been achieved [7], which enriched the structural complexity of nanocarbons and provided various materials applications.

Owning to the development of STM (scanning tunneling microscope) [8,9] and nc-AFM (noncontact atomic force microscope) [10], a wide variety of low-dimensional carbon nanostructures have been synthesized and characterized at the atomic scale on surfaces, such as linear polymers comprising of hydrocarbons [11,12], graphene nanoribbons [13,14,15,16,17,18], porous graphene [19], and polycyclic aromatic hydrocarbons [20,21]. In addition to the all-benzene family with different dimensionality and sizes, novel topological nanostructures with nonbenzenoid rings, including four-/eight-membered rings [22,23] and five-/seven-membered rings [24], have sparked considerable attention owing to extraordinary electronic properties. Compared to traditional wet chemistry, on-surface synthesis is carried out under ultra-high vacuum, which prohibits the contaminants from the surroundings and promotes the formation of atomically precise nanostructures. Insoluble molecules, while are less reactive in solution, can act as precursors on surfaces to synthesize various conjugated low-dimensional nanostructures that are hardly prepared by conventional solution chemistry. Furthermore, metal surfaces could reduce the mobility of molecular precursors to some certain degrees due to their confinement effect, which facilitates the fabrication of unexpected nanostructures. Besides the real-space images obtained from STM and nc-AFM, solid evidence can be characterized by X-ray photoelectron spectroscopy (XPS), which could provide more information about element content and valence. In addition, density functional theory (DFT) is another supplementary tool to support the structural analysis from STM and nc-AFM. The exploration of theoretical investigation could provide possible directions to experimental design, synthetic routes to precursors, and predict the performance and potential applications of novel nanostructures [25].

In contrast to *sp*^2^-hybridization, *sp*-hybridization is of great structural advantages for the linearity and small steric effect of the carbon-carbon bond [26]. The *sp*-hybridized structures, such as cyclo[*n*]carbons and linear carbons, can be cumulenic or polyynic [27]. As sketched in Figure 1, both types of structures have two possible configurations [28,29]. The polyynic form consists of alternating single and triple bonds with different lengths, while the cumulenic form is built from consecutive double bonds. Moreover, linear carbons have been predicted to behave with high tensile strength and superconductivity at room temperature, suggesting potential mechanical and electronic applications [30]. Recently, the properties of carbon atomic wires calculated in carbon nanotubes have been reported, aiming to unravel fascinating behaviors of 1inear carbons such as molecular electronics [31,32]. In particular, graphyne, which contains both *sp*- and *sp*^2^-carbons, is predicted to own an intrinsic bandgap that differentiates from zero-bandgap graphene [33]. As illustrated in Figure 1, 0D cyclo[18]carbon, 1D linear carbon, and 2D graphyne have appealed enthusiasm for their unique theoretical performance in electronic devices and excellent mechanical properties. Experimentally, various novel carbon structures containing *sp*-carbon have been fabricated recently via on-surface synthesis strategy, challenging to traditional methods in solution [34], which can be used to obtain a plethora of pre-designed nanostructures. A lot of theoretical investigations have been employed on carbon clusters to speculate their structures, stabilities, and properties. It is found that odd-numbered carbon clusters have linear structures while most of the even-numbered ones prefer being cyclic forms [35]. Energetically, odd carbon species have lower energies than even ones but even species show greater electron affinity [36]. What is more, atomically precise nanostructures fabricated on the surface could provide possibilities to verify their fascinating theoretical properties. Herein, we will summarize recent works regarding the on-surface synthesis of low-dimensional *sp*-hybridized carbon nanostructures, including cyclo[*n*]carbons, metallic carbynes, and graphdiyne-like structures.

## 2. 0D *sp*-Carbon Nanostructures

0D *sp*-carbon nanostructures were hard to isolate and characterize due to their instability in condensed phases. A large number of experimental and theoretical studies on carbon clusters have been applied to their molecular structures and electronic properties [37]. Most importantly, a fundamental and controversial question is still open: the configuration of cyclocarbons is polyynic (alternating single and triple bonds of different lengths) or cumulenic (consecutive double bonds) [29,38]. Initial laser desorption mass spectrometry has been reported to obtain evidence for the size-selective growth of fullerenes through the coalescence of cyclo[n]carbons, the molecular carbon allotropes consisting of monocyclic rings with a certain number (*n*) of carbon atoms [39]. Three carbon oxides C_n_(CO)_n/3_ (Figure 1a–c) with n = 18, 24, and 30 were prepared as precursors to the cyclocarbons cyclo-C_18_, C_24_, and C_30_ (Figure 1d), respectively. Mass spectroscopy pointed out that the coalescence of cyclo-C_30_ was accompanied by the formation of predominantly buckminsterfullerene (C_60_), while the smaller C_24_O_6_ and C_32_O_8_ preferentially produced fullerene C_70_ instead. This work provides not only new insights into the mechanism of fullerene formation but also a protocol to obtain cyclocarbons. However, the structure of cyclocarbon, whether it is cumulene or polyyne, was not explained. Recently, cyclo[18]carbon (C_18_) was synthesized with atomic precision via atom manipulation and characterized for the first time. By using low-temperature STM-AFM, carbon monoxide was eliminated from the precursor (Figure 1e), a cyclocarbon oxide molecule C_24_O_6_, on a bilayer sodium chloride (NaCl) on Cu(111) at 5 K [40] resulting in the formation of a single C_18_ molecule. The nc-AFM image of the intact precursor was shown in Figure 1f and characterization of C_18_ by high-resolution AFM revealed a polyynic structure of C_18_ with defined positions of alternating triple and single bonds, which was observed as bright lobes under nc-AFM contrast shown in Figure 1g. The AFM contrast evidenced the structure of C_18_ on NaCl with the defined positions of C≡C triple bonds, which is supported by the simulated image in Figure 1h. This work provides the first example of a real-space characterization of cyclo[18]carbon, which shows a polyynic form rather than a cumulenic structure using a combination of experimental and theoretical tools.

Innovative carbon clusters with manifold architectures involving *sp*-carbon can be prepared by using rationally designed halogenated hydrocarbons. For example, a nanostructure involving diyne moieties was fabricated via on-surface assisted homocoupling reaction starting from 1,3-bis(2-bromoethynyl)benzene (shortened as BBEB, Figure 1i) on Au(111) [41]. The submolecular resolution observations confirmed the formation of the organometallic intermediates (Figure 1j) and graphdiyne-like macrocycles after demetallization (Figure 1k). It can be concluded that on-surface synthesis has strong advantages in the preparation of 0D carbon structures. Therefore, more attention should be devoted to the on-surface synthesis and its mechanism for the discovery of novel 0D carbon nanostructures comprising of *sp*-carbons.

## 3. 1D *sp*-Carbon Nanostructures

One-dimensional carbon nanostructures involving *sp*-carbon were less investigated due to the high chemical activity compared to graphene that only consists of *sp*^2^-carbon. Over the recent years, graphdiyne nanowires were gradually developed. Generally, graphdiyne nanowires are termed as *m*-*n* graphdiyne nanowires, where *m* refers to the number of phenyl rings and *n* stands for the number of alkyne moieties in a repeating unit of the backbone [42]. Surface-assisted covalent synthesis, depending on the catalysis [43] and confinement of metal substrates [44], has nowadays been a powerful method to fabricate carbon nanostructures involving *sp*-carbons. For example, Glaser coupling, one of the most popular methods to construct diyne motif, has been widely used in both solution and surface chemistry. However, on-surface dehydrogenation of alkynes always leads to chain branching [45]. Thus, it is very important to fabricate structural motifs involving *sp*-carbon more efficiently. To avoid the side reactions, Zhang et al. chose Ag (877) vicinal surface as the template and fabricated 3,2-graphdiyne nanowires with length up to 30 nm via Glaser homocoupling along the step-edges [46], which is demonstrated in Figure 2a. Moreover, Sun et al. developed a novel path to prepare another motif involving *sp*-carbons, namely cumulene, via dehalogenative homocoupling (Figure 2b). The real-space evidence for the formation of the cumulene motifs indicated by the red arrow in Figure 2e,f was obtained by the nc-AFM [47], which is consistent with the relaxed DFT model in Figure 2c,d, respectively.

In addition to the thermal treatment, Pavliče et al. performed atomic manipulation to trigger programmed dehydrogenation of the precursor on NaCl surface at 5 K [48]. Subsequently, a reductive rearrangement of 1,1-dibromo alkene to polyynes occurred, resulting in the formation of a linear structure with three –C≡C– bonds, as illustrated in Figure 2g. With the help of high-resolution nc-AFM, the authors confirmed the structures of reactant (Figure 2h), intermediate (Figure 2i), and final product (Figure 2j). Aside from the atomic manipulation, Sánchez-Grande et al. reported an on-surface synthesis protocol to produce ethynylene-bridged anthracene polymers on Au(111) (Figure 2k) [49]. First, the deposition of the precursor (11,11,12,12-tetrabromoanthraquinodimethane), a quinoid moiety functionalized with =CBr_2_ (abbreviated as TBAn), on Au (111) giving rise to a close-packed assembly. Further annealing to 400 K enabled debromination as well as efficient diffusion of resulting carbene until the complete coupling and aromatization. With the functional side groups modified in the nanowires, cumulene was transferred into the alkyne (Figure 2i–p), which differs from the previously mentioned cumulene bridges formed via dehalogenative homocoupling of alkenyl gem-dibromides in Sun’s work. Eventually, molecular wires based on ethynylene bridges with a bandgap of 1.5 eV were investigated. The relation between resonant form and topological quantum phase of π-conjugated polymers with increasing size of the acene monomers was further investigated in the following work, which shows an increasing cumulene (=C=C=) character of the bridge in both pentacene and bisanthracene polymers [50]. These works demonstrated the formation of *sp*-carbon moieties through dehalogenation of the functional group of =CBr_2_, and the utilization of *sp*^2^-hybridized units to construct nanoarchitectures consisting of *sp*-carbons.

As previously mentioned, the step-edges of the surfaces can improve the reaction selectivity of terminal alkynes during Glaser coupling. This strategy, however, is restricted due to the accompanied side reactions. To solve the problem, terminal alkynyl bromide groups were introduced on surfaces. Experimentally, dehalogenative C−C coupling reactions were demonstrated to be an effective strategy for introduction of carbon-carbon triple bonds with high efficiency and selectivity. Figure 3a schematically shows the formation of –C–Au–C– organic metallic intermediates (Figure 3b) and diyne products (Figure 3c) [51]. The gold atoms could be released from the organometallic molecular wires at elevated temperatures. Finally, C–C coupled graphdiyne nanowires were synthesized on Au(111). It is conclusive to say that dehalogenative homocoupling of terminal alkynyl bromide groups should be an alternative efficient way for incorporating the acetylenic scaffolding into low-dimensional surface nanostructures. What is more, another strategy was developed to in-situ form C≡C triple bonds on the surface. In Sun’s work, on-surface dehalogenative homocouplings of tribromomethyl-substituted arenes were achieved, directly forming nanowires involving carbon-carbon triple bonds (Figure 3d–f) [52]. In this process, conversion of *sp*^3^-hybridized carbon atoms into *sp*-hybridized ones was successfully realized. Other than the debromination homocoupling, cross-coupling reactions, for example, Sonogashira coupling could be resorted to fabricating nanostructures as well. According to the previous study, Sonogashira coupling has a higher reaction barrier than that of Glaser coupling or trimerization of alkynyl [53]. Therefore, optimization of the reaction conditions appears to be particularly important. Wang et al. reported that the synergy of high temperature, low molecular coverage, and low molecular evaporation rate could reach high selectivity up to 70% for Sonogashira coupling to synthesize graphyne nanowires (Figure 3g). This work not only reports the first construction example of graphyne nanowire via on-surface Sonogashira coupling but also offers deep insight into the underlying mechanism which might be applied to other on-surface cross-coupling reaction systems. Given the successful formation of C≡C triple bonds from *sp*^3^-hybridized carbon, a stepwise reaction tactic was put forward [54]. Further functionalized precursor (1-bromo-4-(tribromomethyl)benzene, shortened as BTBMB) with both the tribromomethyl and aryl bromide groups is shown in Figure 3h, which aims at introducing two types of dehalogenative homocoupling reactions (i.e., C(*sp*^3^)–Br and C(*sp*^2^)–Br). As a result, the graphyne nanowires were synthesized through stepwise debromination followed by homocouplings (Figure 3i,j). Despite the notable mechanical and electrical properties, the application of graphyne/graphdiyne nanowires is still limited because of the difficulty in transferring such structures from metallic substrates to other surfaces.

There exists a linear carbon structure comprising of *sp*-carbon configuration with alternating single and triple bonds of different lengths and non-alternating structures, which are defined as polyyne –C≡C– and cumulene =C=C=C=, respectively. Linear carbon is predicted to possess notable physical properties such as extremely high tensile strength and Young’s modulus, rendering them potential candidates for mechanical applications. What is more, the extraordinary predicted properties such as room temperature superconductivity inspired researchers to explore the synthesis of linear carbon. However, the development of carbyne was restricted by the synthesis of well-defined structures due to its extreme instability. Sun et al. have fabricated the metalated carbyne by on-surface synthesis strategy and characterized by the interplay of the high-resolution STM, nc-AFM, and XPS assisted by DFT calculations [55]. Ethyne was chosen as a precursor molecule, and the experimental conditions were optimized by substrate temperature regulation to avoid undesired side reactions and facilitate the dehydrogenative coupling reaction. Finally, they fabricated the Cu metalated carbyne (with −C≡C−Cu− unit) after deposition of ethyne on the surface held at 450 K (Figure 4a–d). The XPS analysis confirmed the formation of Cu metalated carbyne. The minor peak at 284.3 eV (marked as C1) is attributed to unsaturated carbon atoms on surfaces, and the major peak at 283.2 eV (marked as C2) is the characteristic of carbon atoms bound to copper. Moreover, the semiconducting organometallic polyyne with −C≡C−C≡C−Au− unit has also been fabricated by such a precursor Br_2_C=C=C=CBr_2_ on Au(111) [56]. The skeleton rearrangement from the cumulene moiety to the diyne one (1,4-dibromo-1,3-butadiyne, Br−C≡C−C≡C−Br, shortened as C_4_Br_2_) was confirmed by breaking two C–Br bonds of C_4_Br_4_ via atomic manipulation (Figure 4f). Compared with the traditional solution methods to obtain short carbyne-like structures requiring harsh reaction conditions [57], on-surface synthesis strategy is of great advantage in the formation of long linear carbon structures.

## 4. 2D *sp*-Carbon Nanostructures

A plethora of novel two-dimensional carbon nanostructures are flourishing, such as nanoporous graphene, nonbenzenoid carbon allotropes, etc., especially the nanostructures involving *sp*-hybridized carbons. The C–C triple bond is of great advantage for avoidance of fluctuation arising from *cis*-*trans* isomerization which is different from olefinic bond [58,59]. What is more, the graphyne comprising of *sp*- and *sp*^2^-carbon atoms has been predicted to have a crystalline state, which was also predicted to be one of the most stable carbon phases containing acetylenic groups as a major structural component. Additionally, graphyne was calculated to be a semiconductor with a bandgap of 1.2 eV [60], which demonstrated their possible applications in electronic devices. However, the atomic precise fabrication of graphyne is limited by its high chemical reactivity. Zhang et al. developed a method to grow extended 2D graphdiyne-like networks using terminal alkynes on Ag(111) via homocoupling reaction, as demonstrated in Figure 5a–c [61]. The STM image in Figure 5b revealed a closer inspection of irregular, open-porous networks. The inset showed a magnified image of the associated honeycomb unit, and a corresponding model. The formation of covalent bonds was substantiated by complementary XPS measurements. The absence of the low energy shoulder peak at 283.7 eV (a typical binding energy for methylacetylide) revealed a resulting covalent structure, which is contradictory to that proposed by the organometallic binding mechanism when compared with the simulated XPS spectrum of the organometallic dimer. Other impressive efforts have also been devoted to synthesizing graphyne-like nanostructures. Similarly, Zhang et al. have synthesized highly regular single-layer alkynyl-silver organometallic networks at the micrometer scale via gas-mediated surface reaction (Figure 5d) [62]. Different from the previous strategy through thermal induction combined with catalyzation via substrate, terminal alkyne radicals were obtained via oxygen gas mediated deprotonation on Ag(111), and the activation procedure was confirmed by XPS spectroscopy. The peak at 283.6 eV evidenced the strong interaction between the alkynyl groups and the substrates. There was an absence of the O 1s signature in the XPS spectrum after dosing O_2_, ruling out adsorbed oxygen species and other intermediates containing oxygen. The results indicated that both gas species and substrate play crucial roles in the particular activation process. This work provides a versatile fabrication procedure featuring high chemoselectivity without poisoning the surface. Aside from dehydrogenation, debromination of *sp*- and *sp*^3^-carbon was explored, which greatly avoids the byproducts generated at elevated temperatures. As demonstrated in Figure 5e–g, Xu group applied the on-surface synthesis protocol by introducing dehalogenative homocouplings of alkynyl bromides on Au(111), which results in the formation of an organometallic intermediate and subsequent release of gold atoms at elevated temperature [51]. Furthermore, they synthesized C≡C triple-bonded structural motif by dehalogenative homocouplings of tribromomethyl-substituted arenes, as illustrated in Figure 5h [52]. Notably, the organometallic intermediates were not formed in this process. This work provides a new protocol to convert *sp*^3^-carbon atoms to *sp*-hybridized ones through pre-designated molecular precursor. Moreover, Figure 5i,j demonstrate two potential precursors that could be considered to synthesize graphdiyne and graphyne, respectively.

## 5. Conclusions

In summary, we have reviewed a series of recently reported *sp*-carbon nanostructures through on-surface synthesis: cyclo[18]carbon, metalated carbynes, graphyne/graphdiyne nanowires and graphdiyne-like networks. Moreover, different fabrication procedures to prepare carbon-carbon triple bonds were also discussed in this review: (1) dehydrogenation homocoupling, (2) gas-mediated dehydrogenation organometallic coupling, (3) dehalogenation organometallic coupling, and (4) Sonagashira coupling. Meanwhile, it is still of great necessity to explore new protocols for building up acetylenic scaffolds with high efficiency and selectivity. STM and nc-AFM could be developed to be combined with more advanced *in-situ* characterization techniques to explore reaction pathways systematically, such as tip-enhanced Raman spectroscopy (TERS) and infrared spectroscopy (IR). More novel pre-designed nanostructures could be synthesized on surfaces assisted by DFT. Though various carbon nanostructures could be fabricated by on-surface synthesis strategy, there are still many difficulties in transferring well-defined structures from metal substrates to other surfaces towards molecular devices. Therefore, it is of scientific importance to quest practical use and transfer the carbon nanostructures from the surface in high efficiency.

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
