# Peer review of "On-Surface Synthesis of sp-Carbon Nanostructures"

_nanomaterials, 2021, doi:10.3390/nano12010137_

Round 1
Reviewer 1 Report
The review describes carbon nanostructures. This topic is relevant in many fields of materials science. The manuscript is deserved to be published after revision.
I did not find mentioning of Figure 1 d in the text, as well as figure 2c. Please, check the rest of figures.
Some abbreviations are explained several times in the text. It is sufficient to do only first time.
Please, support this sentence by reference: „Over the past decades, many breakthroughs for the fabrication of novel nanostructures have been achieved, which enriched the structural complexity of nanocarbons and provided various materials application [DOI: 10.3390/MA13102402].“
You have mentioned in the text several time the XPS of the samples. Is it possible to add information about the described structures using this methodology? Because it will show the bonds condition.
Author Response
Response to Reviewer 1 Comments
Point 1: I did not find mentioning of Figure 1d in the text, as well as figure 2c. Please, check the rest of figures.
Response 1: We thank the referee for the suggestions. We have checked and made corresponding revisions.
Action 1:
- Three carbon oxides Cn(CO)n/3 (Figure 1a-c) with n = 18, 24, and 30 were prepared as precursors to the cyclocarbons cyclo-C18, C24, and C30 (Figure 1d), respectively. (P3, L82)
- By using low-temperature STM-AFM, carbon monoxide was eliminated from the precursor (Figure 1e), a cyclocarbon oxide molecule C24O6, on a bilayer sodium chloride (NaCl) on Cu(111) at 5 K [40] resulting in the formation of a single cyclo[18]carbon (C18) molecule. (P4, L99)
- The nc-AFM image of intact precursor was shown in Figure 1f and characterization of cyclo[18]carbon by high-resolution AFM revealed a polyynic structure of C18 with defined positions of alternating triple and single bonds, which was observed as bright lobes under nc-AFM contrast shown in Figure 1g. (P4, L101)
- The real-space evidence for the formation of the cumulene motifs indicated by red arrow in Figure 2e and 2f, was obtained by the nc-AFM [47], which was consistent with the relaxed DFT model in Figure 2c and 2d, respectively. (P4, L128)
- With the functional side groups modified in the nano wires, cumulene was transferred into the alkyne (Figure 2i-p), which differs from the previously mentioned cumulene bridges formed via dehalogenative homocoupling of alkenyl gem-dibromides in Sun’s work. (P4, L139)
- Further functionalized precursor (1-bromo-4-(tribromomethyl)benzene, shortened as BTBMB) with both the tribro-momethyl and aryl bromide groups is shown in Figure 3h, which aims at introducing two types of dehalogenative homocoupling reactions (i.e., C(sp3)–Br and C(sp2)–Br). As a result, the graphyne nanowires were synthesized through stepwise debromination followed by homocouplings (Figure 3i and 3j). (P7, L206)
Point 2: Some abbreviations are explained several times in the text. It is sufficient to do only first time.
Response 2: We have removed the repeated explanations for some abbreviations in the revised version.
Point 3: Please, support this sentence by reference: „Over the past decades, many breakthroughs for the fabrication of novel nanostructures have been achieved, which enriched the structural complexity of nanocarbons and provided various materials application [DOI: 10.3390/MA13102402].“
Response 3: We thank the reviewer for drawing our attention to this publication. We have added this related work into the revised manuscript.
Action 3:
Over the past decades, many breakthroughs for the fabrication of novel nanostructures have been achieved [7], which enriched the structural complexity of nanocarbons and provided various materials application.
- Sobola, D.; Ramazanov, S.; Konecny, M.; Orudzhev, F.; Kaspar, P.; Papez, N.; Knapek, A.; Potocek, M. Complementary SEM-AFM of Swelling Bi-Fe-O Film on HOPG Substrate. MATERIALS 2020, 13, 2402, doi:10.3390/ma13102402.
Point 4: You have mentioned in the text several time the XPS of the samples. Is it possible to add information about the described structures using this methodology? Because it will show the bonds condition.
Response 4: We agree that XPS could provide bond condition and we have added the descriptions in this review.
Action 4:
- We added an XPS result in Figure 4 with caption “(e) C 1s core-level XP spectrum showing the major peak C2 located at a binding energy of 283.2 eV (cyan curve)” and description “The XPS analysis confirmed the formation of Cu metalated carbyne. The minor peak at 284.3 eV (marked as C1) is attributed to unsaturated carbon atoms on surfaces, and the major peak at 283.2 eV (marked as C2) is the characteristic of carbon atoms bound to copper.” (P7, L220)
- P8, L249: The absence of the low energy shoulder peak at 283.7 eV (a typical binding energy for methylacetylide) revealed a resulting covalent structure, which is contradictory to that proposed by the organometallic binding mechanism when compared with the simulated XPS spectrum of the organometallic dimer.
- P9, L268: The peak at 283.6 eV evidenced the strong interaction between the alkynyl groups and the substrates. There was an absence of an O 1s signature in the XPS spectrum after dosing O2, ruling out adsorbed oxygen species and other intermediates containing oxygen.
Response to Reviewer 2 Comments
Point 1: Line 9. scanning probe microscope (SPM) ---> scanning tunneling microscope (STM). At the article, apart from the abstract, “scanning probe microscope” and/or “SPM” do not appear anywhere 

Response 1: We have modified the sentence “With the extensive development of scanning probe microscope (SPM)” to “With the extensive development of scanning tunnelling microscope (STM) and noncontact atomic force microscope (nc-AFM)”.
Point 2: Line 160. GY? GY ---> GDY?
Response 2: We use graphyne and graphdiyne instead of GY and GDY in the revised version.
Action 2:
- Over the recent years, graphdiyne nanowires were gradually developed. Generally, graphdiyne nanowires are termed as m-n graphdiyne nanowires, where m refers to the number of phenyl rings and n stands for the number of alkyne moieties in a repeating unit of the back bone [42] (P4, L118)
- This work not only reports the first construction example of graphyne nanowire via on-surface Sonogashira coupling but also offers deep insight into the underlying mechanism which might be applied to other on-surface cross-coupling reaction systems. (P7, L200)
Point 3: I miss a section (short paragraph) on theoretical research of this type of sp-carbon nanostructures
Response 3: We thank the reviewer for providing the suggestion of complement on theoretical research. We have added the theoretical research on DFT calculations in this review.
Action 3:
- P2, L49: In addition, density functional theory (DFT) is another supplementary tool to support the structural analysis from STM and nc-AFM. The exploration of theoretical investigation could provide possible directions to experimental design, synthetic routes to precursors, and predict the performance and potential applications of novel nanostructures.
- P2, L58: Recently, the properties of carbon atomic wires calculated in carbon nanotubes have been reported, aiming to unravel fascinating behaviors of 1inear carbons such as molecular electronics [31,32]
- P2, L65: A lot of theoretical investigations have been employed on carbon clusters to speculate their structures, stabilities, and properties. It is found that odd-numbered carbon clusters have linear structures while most of the even-numbered ones prefer being cyclic forms [35]. Energetically, odd carbon species have lower energies than even ones but even species show greater electron affinity [36].
Response to Reviewer 3 Comments
Point 1: The manuscript was reviewed 0D, 1D and 2D sp-carbon nanostructures: polycyclic aromatic hydrocarbons i.e. cyclo[18]carbon, graphene, graphyne/graphdiyne like networks. The content of the review was merely too fundamental, and more information related to the characterization studies i.e. XPS, HRTEM, or DFT theories need to be explored in details to support the findings for each 0D, 1D and 2D and sp-sp2-sp3 hybridized carbon allotrope nanomaterials.
Response 1: We thank the reviewer for the suggestions. We have added some related contents as follows:
Action 1:
- P3: a DFT simulated image (Figure1h) in Figure 1 replaced an image not mentioned in previous version, and the caption is “Simulated AFM image based on gas-phase DFT-calculated geometries, corresponding to the difference in probe height in Figure 1g.” (P3, L93) The description is “The AFM contrast evidenced the structure of C18 on NaCl with the defined positions of C≡C triple bonds, which is supported by the simulated image in Figure 1h.” (P4, L101)
- P7, L228: an XPS result in Figure 4 with caption “(e) C 1s core-level XP spectrum showing the major peak C2 located at a binding energy of 283.2 eV (cyan curve)” and description “The XPS analysis confirmed the formation of Cu metalated carbyne. The minor peak at 284.3 eV (marked as C1) is attributed to unsaturated carbon atoms on surfaces, and the major peak at 283.2 eV (marked as C2) is the characteristic of carbon atoms bound to copper.” (P7, L220)
- P8, L249: The absence of the low energy shoulder peak at 283.7 eV (a typical binding energy for methylacetylide) revealed a resulting covalent structure, which is contradictory to that proposed by the organometallic binding mechanism when compared with the simulated XPS spectrum of the organometallic dimer.
- P9, L268: The peak at 283.6 eV evidenced the strong interaction between the alkynyl groups and the substrates. There was an absence of an O 1s signature in the XPS spectrum after dosing O2, ruling out adsorbed oxygen species and other intermediates containing oxygen.
- P9, L287: STM and nc-AFM could be developed to be combined with more advanced in-situ characterization techniques to explore reaction pathways systematically, such as tip-enhanced Raman spectroscopy (TERS) and Infrared Spectroscopy (IR). More novel pre-designed nanostructures could be synthesized on surfaces assisted by DFT.
Point 2: Authors need to explain more the on-surface synthesis method ie. Advantages, disadvantages and challenges faced as compared to other synthesis methods.
Response 2: We thank the reviewer for the suggestions, and we have added the advantages of on-surface synthesis to the introduction part.
Action 2:
P1, L40: Compared to traditional wet chemistry, on-surface synthesis is carried out under ultra-high vacuum, which prohibits the contaminants from the surroundings and promotes the formation of atomically precise nanostructures. Insoluble molecules, while are less reactive in solution, can act as precursors on surfaces to synthesize various conjugated low-dimensional nanostructures that are hardly prepared by conventional solution chemistry. Furthermore, metal surfaces could reduce the mobility of molecular precursors to some certain degrees due to their confinement effect, which facilitates the fabrication of unexpected nanostructures.
Point 3: All the abbreviations i.e. bBEBP, bTBP, d-BEB, C24O6, NaCl, need to be provided and all the abbreviations define format presented in the manuscript were inconsistent i.e. (1-bromo-4-163 (tribromomethyl)benzene, shortened as BTBMB), 4BrAn 120 (11,11,12,12-tetrabromoanthraquinodimethane), alkenyl gem-dibromide molecular precursor (bBVBP). Do not repeat all the definition with abbreviation i.e. high-resolution scanning tunneling microscopy (STM) was repeated many times in the text. Present the definition with abbreviation at the first time and use abbreviation for the subsequent text. Check DFT and all other abbreviations used.
Response 3: We apologize for inconsistent and repeated abbreviations definition. We have removed the repeated definition of STM, DFT, and XPS, leaving the definition only the first time mentioned and provide the full name of NaCl (sodium chloride). Actually, all the abbreviations were kept consistent with their original articles, respectively, and we now keep their consistency in this review.
Action 3: These abbreviations were revised as follows:
1,3-bis(2-bromoethynyl) benzene, shortened as BBEB (P4, L09)
1,1'-biphenyl, 4-(2,2-dibromoethenyl), shortened as BPDBE (P5, L152)
11,11,12,12-tetrabromoanthraquinodimethane, shortened as TBAn (P4, L138&L160)
1,1'-biphenyl, 4,4'-bis(2-bromoethynyl), shortened as BPBBE (P6, L179)
(1-bromo-4-(tribromomethyl)benzene, shortened as BTBMB (P7, L204)
1,3,5-tris-(4-ethynyl phenyl)benzene, shortened as TEPB (P8, L251)
1,4-bis(tribromomethyl)benzene, shortened as bTBP (P6, L185) (distinguished from BTBMB mentioned above)
1,3,5-tris(2-bromoethynyl)benzene, shortened as tBEP (P8, L257) (to be consistent with bTBP)
1,3,5-tris(tribromomethyl)benzene, shortened as tTBP (P8, L259) (to be consistent with bTBP and tBEP).
Point 4: Review more recent related articles found in nanomaterials journals.
Response 4: We thank the reviewer for the suggestion. We have added another related review article published in Nanomaterials as follow:
Action 4:
25. Xu, Z.; Liu, J.; Hou, S.; Wang, Y. Manipulation of Molecular Spin State on Surfaces Studied by Scanning Tunneling Microscopy. Nanomaterials 2020, 10, 2393, doi:10.3390/na
We uploaded an attachment containing point-by-point response to the reviewer’s comments. Please see the attachment.

Reviewer 2 Report
The paper presents research on the surface synthesis of sp-carbon nanostructures. The presentation of methods and scientific results in the current form is satisfactory for publication in the Nanomaterials journal. The minor and significant drawbacks to be addressed can be specified as follows:
- Line 9. scanning probe microscope (SPM) ---> scanning tunneling microscope (STM). At the article, apart from the abstract, “scanning probe microscope” and/or “SPM” do not appear anywhere.
- Line 160. GY? GY ---> GDY?
- I miss a section (short paragraph) on theoretical research of this type of sp-carbon nanostructures.
Author Response

(The authors gave the same response as above.)

Reviewer 3 Report
The manuscript was reviewed 0D, 1D and 2D sp-carbon nanostructures: polycyclic aromatic hydrocarbons i.e. cyclo[18]carbon, graphene, graphyne/graphdiyne like networks. The content of the review was merely too fundamental, and more information related to the characterization studies i.e. XPS, HRTEM, or DFT theories need to be explored in details to support the findings for each 0D, 1D and 2D and sp-sp2-sp3 hybridized carbon allotrope nanomaterials.
Authors need to explain more the on-surface synthesis method ie. Advantages, disadvantages and challenges faced as compared to other synthesis methods
All the abbreviations i.e. bBEBP, bTBP, d-BEB, C24O6, NaCl, need to be provided and all the abbreviations define format presented in the manuscript were inconsistent i.e. (1-bromo-4-163 (tribromomethyl)benzene, shortened as BTBMB), 4BrAn 120 (11,11,12,12-tetrabromoanthraquinodimethane), alkenyl gem-dibromide molecular precursor (bBVBP). Do not repeat all the definition with abbreviation i.e. high-resolution scanning tunneling microscopy (STM) was repeated many times in the text. Present the definition with abbreviation at the first time and use abbreviation for the subsequent text. Check DFT and all other abbreviations used.
Review more recent related articles found in nanomaterials journals.
Author Response

(The authors gave the same response as above.)

Round 2
Reviewer 1 Report
Authors improved the manuscript a lot. Now the paper can be published.
Reviewer 3 Report
The authors had addressed all the comments